# SVSBI: sequence-based virtual screening of biomolecular interactions

Li Shen[1], Hongsong Feng[1], Yuchi Qiu [1] & Guo-Wei Wei [1,2,3]✉

Virtual screening (VS) is a critical technique in understanding biomolecular interactions, particularly in drug design and discovery. However, the accuracy of current VS models heavily relies on three-dimensional (3D) structures obtained through molecular docking, which is often unreliable due to the low accuracy. To address this issue, we introduce a sequence-based virtual screening (SVS) as another generation of VS models that utilize advanced natural language processing (NLP) algorithms and optimized deep $K$-embedding strategies to encode biomolecular interactions without relying on 3D structure-based docking. We demonstrate that SVS outperforms state-of-the-art performance for four regression datasets involving protein-ligand binding, protein-protein, protein-nucleic acid binding, and ligand inhibition of protein-protein interactions and five classification datasets for protein-protein interactions in five biological species. SVS has the potential to transform current practices in drug discovery and protein engineering.

[1] Department of Mathematics, Michigan State University, East Lansing, MI 48824, USA. [2] Department of Electrical and Computer Engineering, Michigan State University, East Lansing, MI 48824, USA. [3] Department of Biochemistry and Molecular Biology, Michigan State University, East Lansing, MI 48824, USA. ✉email: weig@msu.edu

Biomolecules are the building blocks of life and can be classified into various categories including carbohydrates, lipids, nucleic acids, and proteins based on their sizes, structures, physicochemical properties, and/or biological functions. Additionally, the realization of biomolecular functions is often accompanied by direct physical/chemical interactions with other biological molecules, small ligands, ions, and/or cofactors[1]. These interactions highly depend on the three-dimensional (3D) structures and the dynamics of molecules, as well as biomolecular conformational changes, due to their flexibility and allostery. The understanding of biomolecular interactions is the holy grail of biological science.

The last decade has witnessed the rapid advance in computational biology fueled by the achievement of artificial intelligence (AI) and increased computer power. With advanced techniques in data collecting, processing, analyzing, and representing, modern computational biology can study biological processes at extraordinary scales and multiple dimensions. It has achieved great success for various biological tasks[2–4]. The ability to understand biomolecular interactions via advanced AI approaches has a far-reaching significance to a wide range of research fields, including drug discovery[3], virus prevention[5], directed evolution[4], etc. However, the accurate and reliable prediction of biomolecular interactions is still a severe challenge.

Due to the inherently high correlation between structure information and molecular functions, the structure-based approaches achieved high accuracy and reliability in modeling and learning biomolecular interactions[6–11]. As a result, current analysis and prediction of biomolecular interactions rely heavily on the high-quality 3D structures of interactive biomolecular complexes. Unfortunately, experimental determination of 3D structures is both time-consuming and expensive, leading to the scarcity of experimental structures, particularly, the structures of interactive biomolecular complexes. To overcome this difficulty, molecular docking based on searching and scoring algorithms is designed to generate 3D structures of the interactive complexes, such as antibody-antigen complexes and protein–ligand complexes. Molecular docking is widely incorporated in the virtual screening (VS) of biomolecular interactions, offering an alternative means to construct the 3D structures of interactive biomolecular complexes and is a crucial step in computer-aided drug discovery (CADD). However, current molecular docking is prone to mistakes, rendering inaccurate 3D structures and leading to unreliable virtual screening[12]. Despite the breakthrough in (noninteractive single) protein folding prediction by Alphafold2[2], the structure prediction of interactive biomolecular complexes remains a severe challenge. There is a pressing need to develop innovative strategies for the virtual screening of biomolecular interactions.

Alternatively, sequence-based approaches may provide efficient, robust, and easily accessible deep embeddings of biomolecular interactions without invoking 3D structure docking. Sequenced-based approaches are much more widely applicable than structure-based ones because the Genebank has over 240,000,000 sequences, compared to only 200,000 3D protein structures in the Protein Data Bank (PDB), endowing sequence-based approaches much boarder applicability. There are three major types of sequence-based approaches: (1) composition-based methods such as amino acid composition (AAC)[13], nucleic acid composition (NAC)[14], and pseudo AAC (PseAAC)[15]; (2) autocorrelation-based methods such as auto-covariance[16]; and (3) evolution-based methods such as position-specific frequency matrix (PSFM) and position-specific score matrices (PSSM)[15]. Meanwhile, the use of NLP models to analyze the hidden information in molecular sequences, including protein models, has been successful in recent decades[17–19].

Composition-based methods construct embeddings based on the distribution of single residues or substrings. Autocorrelation-based methods are based on statistical measurement of physicochemical properties of each residue, such as hydrophobicity, hydrophilicity, side-chain mass, polarity, solvent-accessible surface area, etc. Evolution-based methods extract the evolutionary information from large databases by evaluating the occurrence of each residue or the score of that residue being mutated to another type. These methods usually outperform composition-based and autocorrelation-based methods due to their efficient use of a large number of molecular sequences selected by billions of years of natural evolution. Natural language processing (NLP) based methods have been widely used to embed molecules. Among them, autoencoders (AE), long short-term memory (LSTM), and Transformer are most popular. A LSTM model, UniRep, provides enables sequence-based rational protein engineering[20]. An in-house autoencoder was trained with 104 million sequences[21]. Evolutionary scale modeling (ESM) is a large-scale Transformer trained on 250 million protein sequences, which achieved state-of-art performance in many tasks, including structure predictions[22]. For DNA in the genome, pre-trained bidirectional encoder representation model DNABERT has achieved success in non-coding DNA tasks, such as the prediction of promoters, splices, and transcription factor binding sites[23]. Furthermore, an in-house small molecular Transformer was trained with over 700 million sequence data[24]. However, none of these methods was designed for biomolecular interactions.

In this work, we proposed a sequence-based visual screening (SVS) of biomolecular interactions that can predict a wide variety of biological interactions at structure-level accuracy without invoking 3D structures. The biological language processing module in SVS consists of multiple NLP models, extracts evolutionary, and contextual information from different biomolecules simultaneously to reconstruct sequence representations for interactive molecules, such as proteins, nucleic acids, and/or small molecules. SVS has a strong generalizability to various types of tasks for biomolecular properties and interactions. In particular, SVS provides the optimal $K$-embedding strategy to study the interactions between multiple (bio)molecules with negligible computational cost. The intramolecular patterns and intermolecular mechanisms can be efficiently captured by our SVS without performing the expensive and time consuming 3D structure-based docking. We showed the cutting-edge performance of SVS on nine prediction tasks, including four binding affinity scoring functions (i.e., protein–ligand, protein–protein, protein–nucleic acid, and ligand inhibition of protein–protein interactions) and five classification datasets for protein–protein interactions (PPIs). Extensive validations indicate that SVS is a general, accurate, robust, and efficient method for the virtual screening of biomolecular interactions.

## Results

**Overview of the SVS framework.** Our SVS is a sequence-based framework offering deep learning predictions of biomolecular interactions (Fig. 1). First, the biomolecular interaction module identifies types of interactive biomolecular partners and treats the problem in the corresponding flow. Then, the related sequences are collected and curated in the biomolecular sequence module. Additionally, the biomolecular language processing module generates the NLP embeddings of individual interactive molecules from their sequence data. Moreover, the $K$-embedding module further engineers interactive $K$-embeddings from individual NLP embeddings to infer their interactive information. Last, the downstream machine learning algorithm module offers the state-of-the-art regression and classification predictions of various biomolecular interactions.

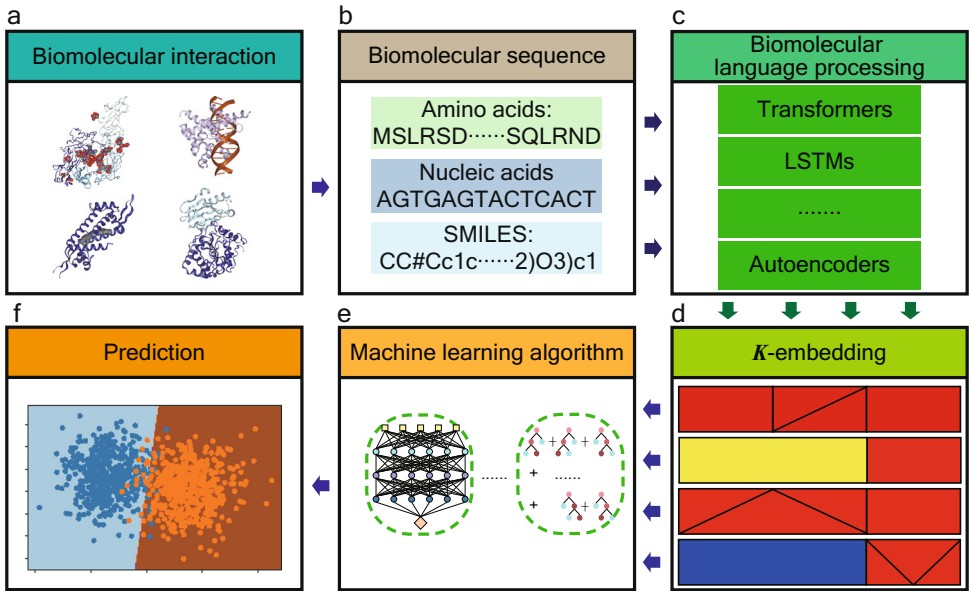

**Fig. 1 Methodological workflow of SVS. a** SVS is designed for a wide variety of biomolecular interactions involving proteins, DNA, RNA, ligands, and their arbitrary combinations. **b** Molecular sequences are extracted from proteins, nucleic acids, and small molecular ligands involved in biomolecular interaction complexes. **c** The biomolecular language processing module presents the NLP embeddings of biomolecular complexes from sequence information. **d** The $K$-embedding module generates the optimal representation of biomolecular interactions from the lower-order embeddings. Each square in the panel represents one kind of 3-embedding strategies. Different patterns represent different 1-embeddings (i.e., an NLP embedding) or a lower-order embedding; different colors represent different integrating functions, which indicate how the $K$-embedding is constructed. **e** Supervised machine learning algorithms learn from the optimal $K$-embedding model of biomolecular interactions. In principle, there are no restrictions on the choice of algorithms. Specifically, in this work, we use GBDT and ANN. **f** Machine learning algorithms are applied to various classification and regression tasks, including membrane protein classifications, therapeutic peptide identifications, protein–protein interaction identifications, binding affinity prediction of protein–protein, protein–ligand, protein–nucleic acids interactions, and inhibition of protein–protein interaction.

In the biological language processing module, NLP embeddings are generated for proteins, nucleic acids, and small molecules using their sequence data (Fig. 1b). We employ various types of NLP models including protein LSTM model (UniRep)[20], protein Transformer (ESM)[22], DNA Transformer (DNABERT)[23], small molecular Transformer[24], and small molecular autoencoder[21]. We particularly focus on Transformer models due to their state-of-art performance with the consideration of sequence dependencies via an attention mechanism[25–27]. Enrich information, such as evolutionary information, 3D structure, and biochemical properties[22,24] can be inferred by Transformers.

The $K$-embedding module ($K$-embedding strategies) takes multiple embeddings from interactive molecular components as inputs and integrates them into an optimal deep $K$-embedding model to decipher biomolecular properties and intermolecular interactions (Fig. 1d). The traditional 3D structure-based virtual screening models require a molecular docking procedure to generate the 3D molecular structures of the interactive complexes, which is inefficient and unreliable[28]. The accuracy and effectiveness of a structure-based docking method are jointly determined by multiple sub-processes including molecular structure determination[1], rigid and flexible docking space search[1], and scoring function construction[29]. Current studies have achieved success in each of these sub-processes. However, minor errors in these sub-processes may accumulate and result in unreliable structure-based docking. Alternatively, in our SVS framework, the $K$-embedding strategies can convert the distribution information of interactive molecular embeddings into the optimal $K$-embedding and extract essential characteristics of biomolecular interactions, which enhances the modelability of machine learning algorithms in learning hidden nonlinear molecular interactive information.

The machine learning module takes the $K$-embedding strategies from the $K$-embedding module for molecular property predictions. The downstream machine learning algorithms include artificial neural network (ANN) and gradient boost decision tree (GBDT) for predictive tasks. The hyperparameters of both models are systematically optimized via Bayesian optimization or grid search to accommodate for different sizes of datasets and deep $K$-embeddings, and different tasks (Machine learning algorithms and Bayesian optimization for ANN hyperparameter tuning). For each task, the optimal $K$-embedding strategy is chosen with the above optimization hyperparameters which achieve the best predictive score in accuracy for classification or in the Pearson correlation coefficient for regression.

**Biomolecular binding affinity predictions**. Quantitatively, binding affinity, defined as the strength of molecular interactions, is reflected in the physicochemical terms of dissociation constant ($K_d$), inhibitor constants ($K_i$), half maximal inhibitory concentration ($IC_{50}$), or corresponding Gibbs free energy[30]. Accurate predictions of molecular binding affinities are not only an important step in modeling biological systems but also a fundamental issue for several practical usages including drug discovery[8,10,31], molecular engineering, and mutagenesis analysis[4].

**Biomolecular binding affinity predictions: protein–ligand binding scoring**. The scoring of protein–ligand binding complexes is the ultimate goal of virtual screening in drug discovery. Typically, millions of drug candidates are screened for a given drug target. The accuracy and efficiency of virtual screening are essential for drug discovery[8,32]. Currently, inaccurate 3D

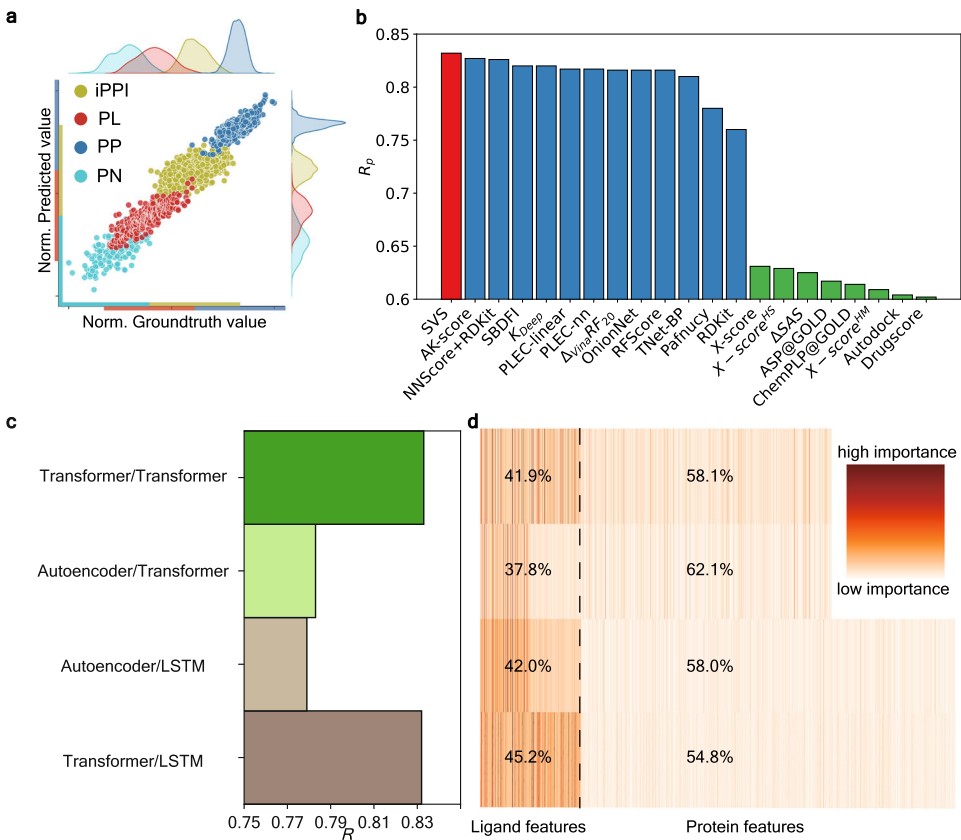

**Fig. 2 Performance analysis of SVS for biomolecular binding affinity predictions. a** A comparison of scaled predicted binding affinities and experimental results for the binding affinity predictions of protein–ligand (PL), protein–nucleic acid (PN), protein–protein (PP), and the inhibition of PPI (iPPI) datasets. Each dataset is scaled to a specific region with an equal range for clear visualization. **b** Comparison of the Pearson correlation coefficient ($R_p$) of our SVS model and that of other structure-based approaches for the protein–ligand binding affinity prediction of the PDBbind-2016 core set[33]. Results in red, blue, and green colors are obtained using no structure (i.e., sequence), experimental structures, and docking generated structures of protein–ligand complexes, respectively. Our SVS outperforms the state-of-the-art models, such as AK-score[7], NNScore+RDKit[38], and many others[9,33-37]. **c** Comparison of different NLP models for the Pearson correlation coefficients $R_p$ of the protein–ligand binding prediction. **d** The relative importance distributions of different NLP models as shown in **c**. Each row consists of 512+1280/1900 colored vertical line, and each represents the importance of one feature that is generated by the NLP models. The black dashed line is the dividing line for features belonging to different type of molecules. The percentage on the left or the right of the black dashed line is the proportion of the summation of importance of features for the same type of molecules.

structure-based docking and the associated unreliable virtual screening are the main obstacles in rational drug design and discovery.

In this study, we applied SVS to predict the protein–ligand binding affinity on the PDBbind 2016 dataset[33], a popular benchmark dataset employed by hundreds of research teams to validate their protein–ligand binding scoring functions[7–9,33,33–38]. It has the training data of 3772 protein–ligand complexes from the PDBbind 2016 refined set and the test data of 285 complexes from the core set. The availability of 3D complex structures in PDBbind database favors structure-based scoring functions, such as algebraic topology-based machine learning models, such as TopBP[10], PerSpect-ML[31], and AA-score[32].

The best performance of 2D fingerprint-based methods, achieved by the protein–ligand extended connectivity (PLEC) fingerprint[35], was $R_p = 0.817$. In fact, 3D structure information was utilized in PLEC, highlighting the importance of 3D structures in existing protein–ligand binding scoring functions. We select this dataset to examine whether the proposed SVS, without resorting to structural information, can reach the same level of accuracy as structure-based scoring functions.

As shown in Fig. 2b, our SVS model gives the accurate prediction of binding affinity with $R_p = 0.832$ and RMSE 1.696 kcal mol$^{-1}$ (Fig. 2b). For structure-based methods, $R_p > 0.7$ can

be usually achieved if experimental structures of protein–ligand complexes are used, while lower $R_p < 0.65$ is achieved when molecular docking, such as ASP@GOLD and Autodock, is used to generate the structures of protein–ligand complexes[33]. The structure-based TopBP method, using algebraic topology to simplify the structure complexity of 3D protein–ligand complexes, achieved the best performance with $R_p$/RMSE of 0.861/1.65 kcal mol$^{-1}$[10] in the literature. Excluding advanced mathematics-driven structure-based methods, SVS outperforms other structure-based methods, e.g., AK-score[7] ($R_p$: 0.827), NNScore+RDKit[38] ($R_p$: 0.826) (Fig. 2b). This achievement is of enormous significance that the quality and reliability of the current virtual screening can be dramatically improved to the level of x-ray crystal structure-based approaches without depending on 3D experimental structures. Our result has a far-reaching implication—reliable virtual screening can be carried out on any drug target without relying on the 3D structures of drug–protein complexes.

The performance from different combinations of protein and ligand embeddings are further explored (Fig. 2c). We used ESM Transformer[22] and UniRep LSTM[20] model for protein embedding, and a Transformer[24] and an autoencoder[21] model for ligand embedding. Our analysis indicates that the small molecular Transformer outperforms the autoencoder. Additionally,

Transformer achieves better performance than LSTM model for protein embedding. Further feature analysis is provided from the feature importance analysis from GBDT (Fig. 2d). Both small molecular embeddings have the dimension of 512. For the protein embeddings, Transformer dimension is 1280, and LSTM is 1900. First, small molecular features have more highly important ones. The average importance of small molecular features are 0.082 (41.9/512), 0.074, 0.082, and 0.088 for four cases from top to bottom (Fig. 2d). In contrast, the average importance of protein features are 0.045, 0.049, 0.031, and 0.028 for four cases. Additionally, the small molecular Transformer offers more important features than the autoencoder does. For the protein embeddings, the Transformer has more important features than the LSTM does. Therefore, the combination of the ligand Transformer and protein ESM Transformer achieves the best prediction as shown in Fig. 2c.

**Biomolecular binding affinity predictions: protein–protein binding affinity prediction**. Protein–protein binding affinity refers to the strength of the attractive interaction between two proteins, such as an antibody–antigen complex, when they bind to each other. It is important metric for assessing the stability and specificity of protein–protein interactions (PPIs), which are vital for many biological processes.

Understanding protein–protein binding affinity is important for many applications, including drug discovery, antibody design, protein engineering, and molecular biology. For example, knowing how antibody-antigen binding affinity is affected by the shape of the antibody, the charge and hydration of the antibody, and the presence of specific binding sites or residues on the antibody, one can engineer antibodies with specific binding properties to neutralize viruses[39,40].

The protein–protein binding affinity can be quantified by Gibbs free energies. The surface plasmon resonance (SPR), isothermal titration calorimetry (ITC), enzyme-linked immunosorbent assay (ELISA), and Western blotting are used to determine protein–protein binding affinities. In our work, we build a SVS model to predict protein–protein binding affinities from protein sequences. We collect and curate a set of 1695 PPI complexes (Datasets) in the PDBbind database[41]. This dataset is employed to show the versatile nature of SVS. Sequences of these PPI complexes are extracted and embedded using the Transformer. The PPIs are represented by the stack of their Transformer embeddings in our study. Our SVS model reached the $R_p$ of 0.743 and the RMSE of 1.219 kcal mol$^{-1}$ via 10-fold cross-validation, and the comparison of predicted value versus the ground truth is shown in Fig. 2a. Our result indicates SVS is a robust approach for predicting the binding affinity of PPIs.

**Biomolecular binding affinity predictions: protein–nucleic acid binding affinity prediction**. Another class of biomolecular interactions is protein–nucleic acid binding which plays important roles in the structure and function of cells, including catalyzing chemical reactions, transporting molecules, signal transduction, transcription, and translation. It is also involved in the regulation of gene expression and in the maintenance of chromosome structure and function. Dysregulation of protein–nucleic acid binding can lead to various diseases and disorders, such as cancer, genetic disorders, and autoimmune diseases. The understanding of the factors, such as hydrogen bonding, dipole, electrostatics, Van der Waals interaction, hydrophobicity, etc. that influence protein–nucleic acid binding affinities can be utilized to design new therapeutic molecules.

In this work, we apply SVS to analyze and predict protein–nucleic acid binding affinity. Due to the lack of existing benchmark datasets, we extract a dataset from the PDBbind database[41]. A total of 186 protein–nucleic acid complexes was collected (Datasets). This dataset is chosen to demonstrate that the SVS works well for predicting nucleic acid-involved biomolecular interactions. For this problem, our SVS utilizes a Transformer (ESM) for embedding protein sequences and another Transformer (DNABERT) for embedding nucleic acid sequences. Our model shows good performance with an average $R_p$/RMSE of 0.669/1.45 kcal mol$^{-1}$ in a 10-fold cross-validation. Our results are depicted in Fig. 2a. Considering the fact that the dataset is very small, our SVS prediction is very good.

**Biomolecular binding affinity predictions: inhibition of protein–protein interaction prediction**. Having demonstrated SVS for protein–ligand, protein–protein, protein–nucleic acid binding predictions, we further consider a problem involving multiple molecular components. The small molecule inhibition of protein–protein interaction prediction (iPPI) involves at least three molecules.

Protein–protein interactions are essential for living organisms. Dysfunction of PPIs can lead to various diseases, including immunodeficiency, autoimmune disorder, allergy, drug addiction, and cancer[42]. Therefore, the inhibition of PPIs (iPPIs) is of great interest in drug design and discovery. Recent studies have demonstrated substantial biomedical potential for iPPIs with ligands[43].

However, iPPI with ligands is challenging in a vast range of investigation phases including target validation, ligand screening, and lead optimization[44]. Traditional computational methods for iPPI predictions have various limitations. For example, structure-based approaches have to overcome the complexity of ligand docking caused by the large and dynamic interfaces of PPIs even with stable and reliable experimental complex structures[45]. Recently, Rodrigues et al.[42] have developed an interaction-specific model, called pdCSM-PPI, which utilizes graph-based representations of ligand structures in the framework of ligand-based virtual screening. An important characteristic of their approach is that their models are ligand-based and target-specific: the input of each model is a set of ligands that target one particular PPI. Instead of exploring the hidden mechanism of iPPI, their models rely on a comparison of ligands by assuming that ligands with similar structures exhibit similar behavior, i.e., the similar property principle. Their approach avoids the difficulties of lacking iPPI structures and molecular mechanisms by using target-specific predictions, in which one machine-learning model is built for ligands targeting the same PPI system. Therefore, it cannot be used for the screening of new targets. By contrast, SVS can avoid this difficulty by sequence embedding of PPI targets. As a result, SVS can be directly applied to explore the inhibition of new PPIs without matching targets in existing iPPI datasets.

In this work, we analyzed PPIs and ligands by using various $K$-embedding strategies, to predict the half-maximal inhibitor concentration (IC50) of the ligand inhibition of PPI. For each iPPI complex, a small molecular Transformer and a protein Transformer are used to embed one ligand sequence and two protein sequences in our SVS. We tested our model on the dataset considered by Rodrigues et al.[42] Our model shows an $R_p$ of 0.766 and RMSE of 0.761 mol/L in the 10-fold cross-validation, while the $R_p$ and RMSE of the earlier pdCSM-PPI model are 0.74 and 0.95 mol/L, respectively. SVS shows a better performance in both $R_p$ and RMSE, illustrating the superiority of the SVS method. The comparison of predictive results versus the ground-truth value of our model can be found in Fig. 2a.

We explore $K$-embedding strategies via various NLP deep embeddings. We examine three integrating functions in this

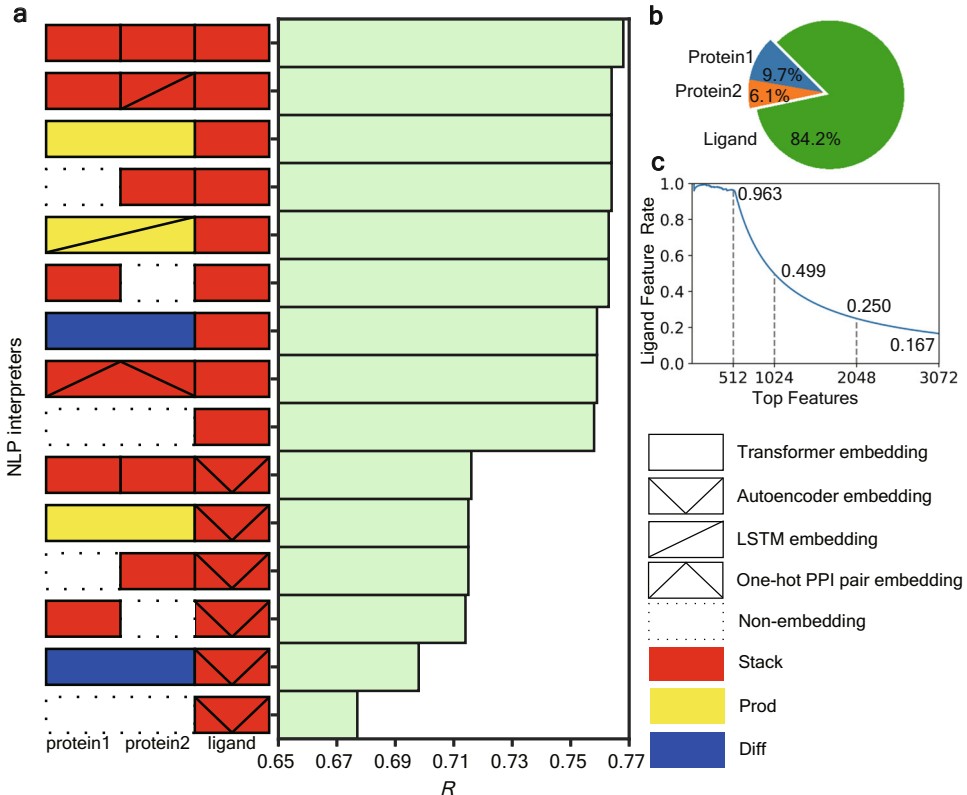

**Fig. 3 Performance analysis of various *K*-embeddings for iPPI dataset. a** Illustration of the performances (*R*ₚ) of various *K*-embedding strategies. **b** The feature importance analysis of ligand, protein1, and protein2 in iPPI predictions using the best *K*-embedding strategy (i.e., the stack of three Transformers). **c** The proportion of ligand features in top features of SVS for iPPI using the best *K*-embedding strategy (i.e., the stack of three Transformers). The *x*-axis indicates the quantity of top features to be considered and the *y*-axis represents the proportion of ligand features in the top features.

study, i.e., Stack, Prod, and Diff, to generate *K*-embedding strategies with the higher-order embedding built from lower-order embeddings. Stack concatenates two biomolecular language processing embeddings from two proteins in a PPI complex into a single embedding vector. This method preserves the complete information provided by the biomolecular language processing module, but the downside is its high dimensionality. Since two proteins in a PPI complex are encoded by two vectors of identical length, 2-embedding can be done via the component-wise operations between these two vectors. We also tested the component-wise product (Prod) and the absolute value of the difference (Diff). These component-wise 2-embedding approaches result in lower-dimensional 2-embeddings for the downstream machine-learning module. The specific formulas corresponding to these three strategies are described in Eqs. (2), (3), and (4), respectively.

Here, we choose 14 kinds of higher-order deep embeddings that take the full consideration of the homogeneity or heterogeneity of NLP models, which are shown in Fig. 3a with their predictive performance. It is worth noting that this iPPI dataset is a ligand-central dataset consisting of multiple ligands that target the same PPI. Therefore, 1-embedding for ligand sequence information processing will play the most important role. Our experiments show that using Transformer-based models with the Stack schemes will give a state-of-the-art performance.

We further analyze the feature importance of our best schemes from GBDT for features encoding ligands and proteins. Interestingly, features for ligands are substantially more important than that for proteins (Fig. 3b). Specifically, the importance for ligand features is much higher at 84.2%, while the sum of importance for two proteins is only 15.8%. On the other hand,

top features include a high proportion of ligand features, for example, 96.4% of the top 512 features are from ligand features (Fig. 3c). A possible reason for such feature imbalance may be because only a few PPI systems are included in this dataset which has 1694 ligands but only 31 PPIs. Despite protein features being less important, they are necessary for learning iPPI without matching targets. As shown in Fig. 3a, without PPI information (non-encoding of PPIs), or with only trivial classification information of PPI (one-hot pair encoding of PPIs), our models show a substantial decline in the predictive accuracy. The only exception is Diff of the PPI target. One reason is that many proteins in this PPI target belong to the same protein family. Thus, the high similarity of these proteins in sequence would only provide very limited information for Diff schemes. In general, the protein features are necessary components for learning target-unmatched iPPIs.

**Protein–protein interaction identification.** Protein–protein interactions (PPIs) regulate many biological processes, including signal transduction, immune response, and cellular organization[46]. However, the selectivity and strength of PPIs depend on species and the cellular environment. Identifying and studying PPIs can help researchers understand the molecular mechanism of protein functions and how proteins interact with one another within a cell or organism.

We utilized the SVS method to identify PPIs, where our model classified protein pairs in a given dataset following standard training and test splitting protocols in the literature[14,47]. Positive samples were defined as interacting protein pairs that are in direct physical contact through intermolecular forces, while negative

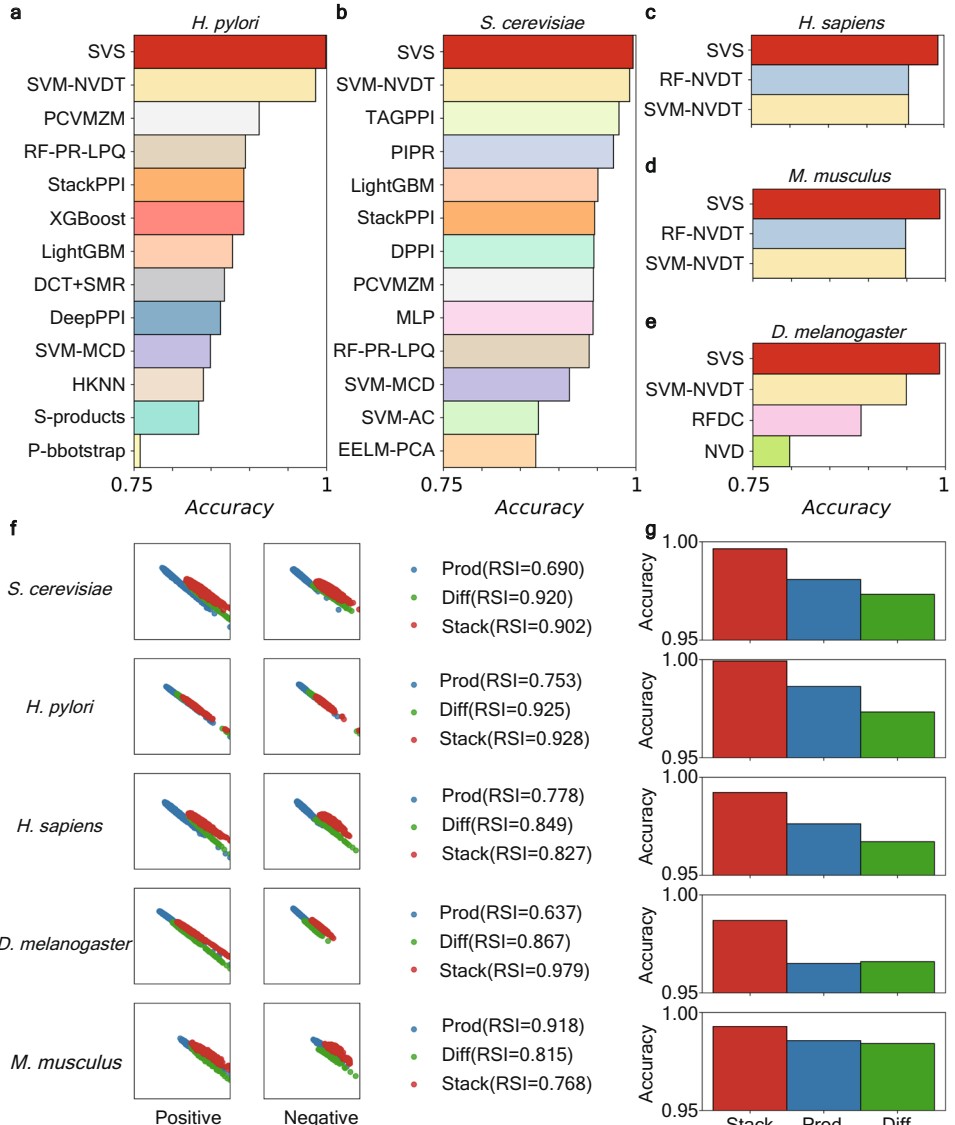

**Fig. 4 Performance analysis of the SVS for five protein–protein interaction and non-interaction classification datasets. a–e** Comparisons of our predictive model (SVS) with some previous PPI identification models. The comparison of each dataset is shown independently in a subplot with the name of the dataset at top of it. For each subplot, the x-axis represents the accuracy scores, ranging from 0.75 to 1; the y-axis lists the name of each model. Our SVS outperforms the state-of-art models, such as SVM-NVDT[14], RF-NVDT[14], PCVMZM[58], TAGPPI[47], etc. **f** Comparison of different K-embedding strategies, measured by R-S analysis on features. Three K-embedding strategies, Prod, Diff, and Stack, are chosen for comparison. This plot is vertically composed of five similar sections. Each section represents a dataset with the name on the left. Furthermore, each section possesses two parts. The left part has two subplots showing the R-S plot of positive or negative features generated by different strategies. The right part shows the R-S Index (RSI) of different strategies. **g** The comparison accuracy of predictive models of different K-embedding strategies.

samples were generated by randomly selecting protein pairs in distinct sub-cellular compartments[14,47]. Five PPI datasets with different species including *Homo sapiens* (HS), *Mus musculus* (MM), *Saccharomyces cerevisiae* (SC), *Drosophila melanogaster* (DM), and *Helicobacter pylori* (HP) are employed for the benchmark. Here, we explore three K-embedding strategies: Stack, Prod, and Diff.

Since the performance of regression models is complicated, we first analyze the performance of interactive features without downstream regression models. In particular, we employed the R-S plot to visualize feature residue score (R) versus similarity score (S)[48]. The R-score and S-score of a given sample are calculated by considering the distances of its features with that of inter-class samples and intra-class samples, formulated as Eqs. (10) and (11), respectively. Both R-score and S-score range from 0 to 1. A

sample with a higher R-score indicates that it is far from samples in other classes, and a higher S-score indicates that it is close to other samples in the same class. An effective featurization method is expected to have both high R-scores and S-scores, despite a clear trade-off exists between R- and S-scores (Fig. 4b). Notably, such a trade-off can also be quantified by the R-S index (Eq. (14)). The R-S analysis shows that Stack features are located at the upper right of Prod and Diff embeddings except for the *H. pylori* dataset (located in a similar area), though they overlap extensively over all datasets. In addition, from the perspective of the R-S index, Stack and Diff have advantages in two datasets, and Prod has advantages in one dataset.

Furthermore, we compared different K-embedding strategies by coupling with the identical regression models using fivefold cross-validation (Fig. 4b). Consistently, the Stack strategy showed

the highest accuracy score than others in their downstream model performance for all datasets tested (Fig. 4c). Overall, Stack provides an optimal $K$-embedding strategy.

Overall, our models with the best *Stack* of biomolecular language processing embeddings showed accuracy scores as high as 99.93%, 99.28%, 99.64%, 99.22%, and 98.69% for datasets *Helicobacter pylori*, *Mus musculus*, *Saccharomyces cerevisiae*, *Helicobacter pylori*, and *Drosophila melanogaster*, respectively (Fig. 4a and Supplementary Table 1). In comparison, the state-of-art method, SVM-NVDT[14], gives 98.56%, 94.83% 99.20%, 95.41%, and 94.94%, respectively for these datasets. SVM-NVDT was based on natural vectors and dinucleotide and triplet nucleotide information. Also, the Supplementary Note 2 displays additional results of our SVS models including the AUC curves which are shown in the Supplementary Fig. 1. Our models outperform all previous models by a substantial margin, which demonstrates the superiority of our method over previous methods for identifying PPIs.

## Discussion

In this study, we utilize representations from traditional molecular language models as a starting point to inductively define high-order $K$-embeddings, which provide a systematic strategy for representing biological interactions involving an arbitrary number of molecules. By generating different $K$-embeddings, we can effectively and easily capture the sequence representations of NLP models generated for a single molecule. These $K$-embeddings allow for comprehensive consideration of the potential heterogeneity of interactive biomolecules, enhancing the representability of individual molecules. Furthermore, the design of $K$-embedding enables SVS to optimize downstream machine/deep learning algorithms. To demonstrate the utility of $K$-embeddings, we design two machine learning algorithms that achieve state-of-the-art results.

In predicting biomolecular interactions, structure-based approaches are popular and highly accurate when the topological representations of high-quality 3D structures are employed[10]. However, their performance depends on the availability of reliable high-resolution experimental structures. Structural docking is a necessary protocol for structure-based approaches when there is no experimental structure available for the interactive complex. Additionally, the power of structure-based methods lies in their ability to accurately capture the geometric information of the interactive complexes. Therefore, the disparity between docked structures and experimental structures will also be inherited by structure-based models. However, no studies have shown that current molecular docking models can control this disparity within acceptable tolerances. By contrast, our SVS method provides an alternative approach for the study of interactive molecular complexes using only sequence data. It implicitly embeds structural information, flexibility, structural evolution, and diversity in the latent space, which is optimized for downstream models through $K$-embedding strategies. It is worth noting that SVS reaches the same level accuracy as of the best structure-based approach as shown in Fig. 2.

Ligand-based virtual screening models also serve as another effective approach that can avoid structure-based docking for evaluating biomolecular interaction with ligands[49]. However, the current usage of ligand-based models is quite limited as these models in principle can only be applied to target-specific datasets and cannot be used for the screening involving new targets. We showed that by combining target and ligand deep embeddings via $K$-embedding strategies, SVS gives rise to robust target-unspecific predictions with structure-based accuracy.

The Biological language processing module and the $K$-embedding module are two major components in SVS models.

Conventionally, the model performance relies on both featurization modules and machine learning algorithms. To solely analyze the quality of the featurization modules, we carry out residue-similarity ($R$-$S$) analysis using $R$-$S$ plot and $R$-$S$ index[48] for classification tasks (Fig. 4b). The $R$-$S$ analysis describes the quality of features in terms of similarity scores and residue scores as well as the deviation between different classes.

We further analyze SVS behaviors on different datasets in terms of magnitudes and modelability (Fig. 5a) where the basic information of correspondence datasets can be found in Supplementary Table 3. Three metrics are employed: modelability index, predictive, and index magnitude index. The modelability index and magnitude index are calculated based on the training data of each dataset, while the predictive index is calculated based on our predictive results on the test data. Note that if our model is tested via cross-validation, then the whole dataset will be calculated for each of the five indices. The predictive index is chosen based on task types: we chose the accuracy score for classification tasks and $R_p$ for regression tasks. The modelability index, which represents the feasibility of our approach on the training data of each dataset, is evaluated by calculating the class-weighted ratio (classification) or the activity cliff (regression) between the nearest-neighbors of samples (Eqs. (15) and (16)). Previous studies[50,51] have suggested that 0.65 is the threshold to separate the modelable and non-modelable datasets. Our model exceeds this threshold in all datasets. In particular, the modelability indices exceed 0.8, which confirms the robustness, stability, and feasibility of our SVS. Our method is compatible with a wide variety of dataset sizes, as shown by the magnitude index, which reflects the corresponding dataset size in proportion to the maximal size of the 9 datasets studied (the maximal data size is 11,188). Our analysis shows that there is no substantial correlation between the magnitude index and with modelability index or the predictive index, with the only exception being the PN dataset. This dataset, compared to other datasets of the same task (i.e., PL, PP, iPPI datasets), has the same level of modelability index, but with lower levels of the predictive index. We believe that this is because the magnitude index is too small, and this dataset is tested by cross-validation. Therefore, the randomly selected data leads to a void in the feature space, making it difficult for our model to fit this dataset. In conclusion, SVS can be broadly applied for biomolecular predictions and is robust against data size variation. Moreover, SVS has a strong adaptability to molecules with different sequence compositions. Since proteins were involved in each of our previous numerical experiments, we show the length distribution of protein sequences in each dataset (Fig. 5b) as well as the distribution of amino acids appearance rate in the sequences (Fig. 5c). On average, the sequence lengths of PL, PP, and PN are shorter than those of *Saccharomyces cerevisiae* (SC), *Drosophila melanogaster* (DM), *Helicobacter pylori* (HP), *Homo sapiens* (HS), and *Mus musculus* (MM). This is because samples in the previous datasets are also provided with experimentally determined structures. The availability and reliability of large-size protein structures are subjected to experimental techniques as well as practical considerations, which leads to inevitable systematical bias for structure-based approaches. On the other hand, our SVS models show excellent performances for tasks involving various sequence length distributions. Furthermore, the diversity of the amino acid appearance rate distribution supports the adaptability of our model for tackling different biological tasks, regardless of whether the sequence composition involved has some specificity. In conclusion, our SVS models are robust against sequence length variation and adaptive to biomolecular variability, which reveals the potential of our SVS method as a universal approach for studying biological interactions.

The success of the SVS is due to the use of powerful NLP models, such as LSTM, autoencoder, and particularly Transformers trained

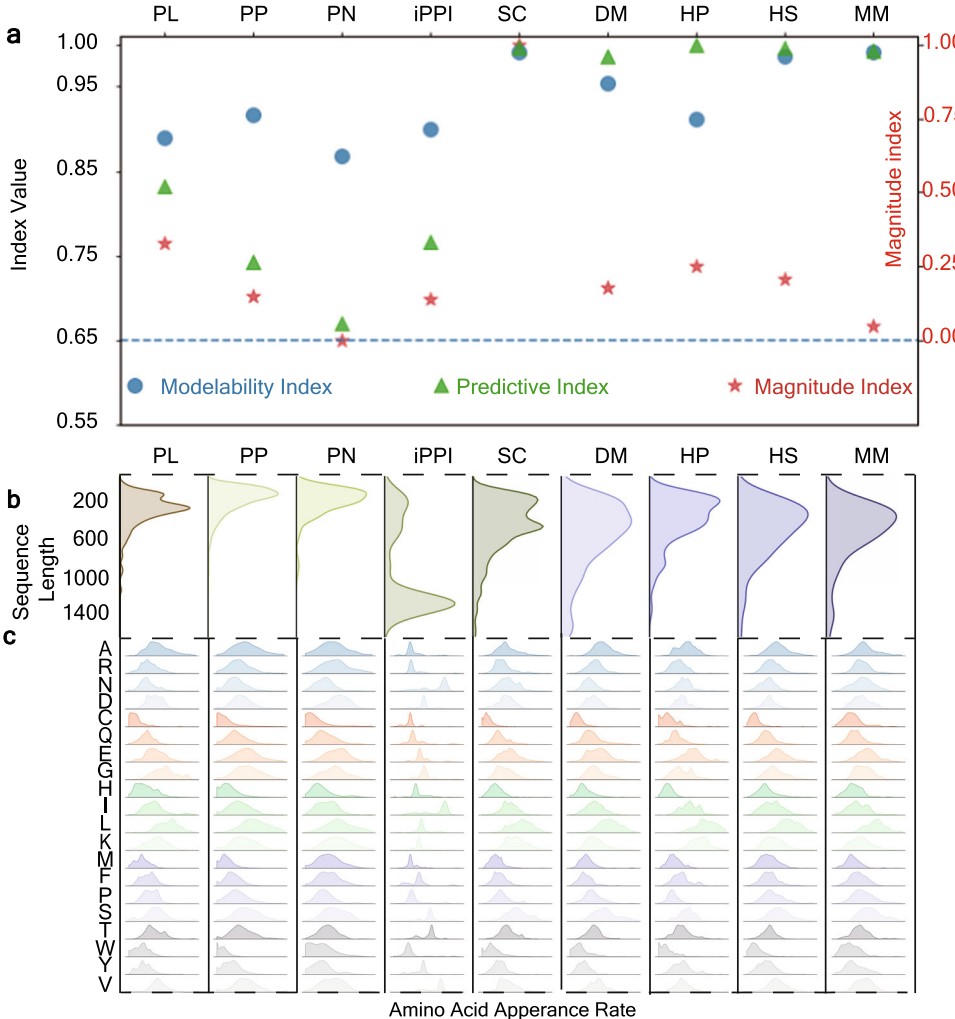

**Fig. 5 Analysis of nine datasets. a** Modelability index, predictive index, and magnitude index for nine datasets. The left *y*-axis represents modelability and predictive indices, while the right *y*-axis is the magnitude index. Nine datasets used in our work are four binding affinity regression tasks (i.e., PL, PP, PN, iPPI), and five protein–protein interaction classification tasks, namely SC (*Saccharomyces cerevisiae*), DM (*Drosophila melanogaster*), HP (*Helicobacter pylori*), HS (*Homo sapiens*), and MM (*Mus musculus*). **b** The distribution of sequence length for 9 datasets. **c** The normalized amino acids appearance rate distribution. This subfigure has nine channels horizontally, corresponding to nine datasets described in **a**, **b**. Each channel shows the distribution of 20 types of amino acids appearance rates in sequences of the dataset.

with hundreds of millions of molecules. These models extract the constitutional rules of molecules and biomolecules without resorting to molecular property labels. The proposed SVS will become more powerful as more advanced NLP models become available.

To showcase the proposed SVS method, we choose nine representative biomolecular interaction datasets involving four regression datasets for protein–ligand binding, protein–protein binding, nucleic acid binding, and ligand inhibition of protein–protein interactions and five classification datasets for the protein–protein interactions in five biological species. SVS can be applied to the large-scale virtual screening of multiple targets and multiple molecular components without any structural information.

Recently, there has been a growing concern about possible data leakage in machine learning models, where the model may rely too heavily on sequence similarity to make predictions[52]. This issue undermines the ability of the model to learn the underlying pattern of interactions among biomolecules. However, our approach, SVS, avoids data leakage by utilizing NLP-based *K*-embeddings. By extracting a wide range of hidden information from sequences, including structure, contextual, biochemical, and evolutionary information, our SVS model is less dependent on

sequence similarity. Recent studies also demonstrate the effectiveness of NLP-based methods in predicting single or multiple mutations of protein interactions that may completely alter or abandon molecular interactions[4,53], further confirming the low dependence of SVS on sequence similarity.

## Methods

**Datasets**. In this study, we used PDBbind-2016 datasets[41] for predicting the protein–ligand binding affinity. The dataset used in protein–protein binding affinity was constructed from PDBbind database[41]. The original PDBbind version 2020 contains binding affinity data of 2852 protein–protein complexes. We selected 1695 samples with only two different sub-chain sequences as shown in Supplementary Table 5. Furthermore, we also construct the protein–nucleic acid binding affinity dataset from PDBbind version 2020. However, unlike proteins and ligands, nucleic acids need to be converted to *k*-mers (in our models, *k* equals 3) before feeding into the Transformer model we used. Thus, one unconventional letter (e.g., X, Y) in a sequence will result in *k* unknown *k*-mers. In addition, nucleic acids binding to proteins are generally short in length. Therefore, thus unconventional letters in their sequences may completely destroy the context of *k*-mer representations. For example, a nucleic acid sequence "ACXTG" will be converted into three 3-mers: "ACX", "CXT", and "XTG". Note that these three 3-mers all contain an "X", so the biomolecular language processing model will treat them as unknown tokens, and will not be able to read any useful sequence information. In order to guarantee the effectiveness of sequence information, we apply a stricter excluded criterion: 1)

exclude those protein–nucleic acid complexes that their sequence numbers do not equal two; 2) exclude those protein–nucleic acid complexes that have unclear labels; 3) exclude those protein–nucleic acid complexes that have abnormal letters (normal ones are A, C, T, G) in its nucleic-acid sequences; 4) exclude those protein–nucleic complexes that whose nucleic acid sequence length is fewer than 6. The resulting dataset contains 186 protein–nucleic acid complexes as shown in Supplementary Table 4. Additionally, for these two datasets, the labels are transformed from dissociation constant ($K_d$), inhibitor constant ($K_i$), and half maximal inhibitory concentration ($IC_{50}$) to Gibbs free energy based on the Supplementary Eq. 8.

The original dataset iPPI dataset focuses on ligands thus the availability of PPI targets is obscure and only 31 targets are provided at the family level while 1694 ligands are available. For each protein family, we selected one protein to represent the whole family (e.g., we chose P10415/Q07812 for BCL2/BAK; O60885/P62805 for bromodomain/histone, and O75475/P12497 for ledgf/in.). More specific correspondences can be found in Supplementary Table 6.

The protein–protein interaction identification involves five benchmark datasets, namely, 2434 proteins pairs from *Homo sapiens*, 694 protein pairs from *Mus musculus*, 11,188 protein pairs from *Saccharomyces cerevisiae*, 2140 protein pairs from *Drosophila melanogaster*, and 2916 protein pairs from *Helicobacter pylori*[14]. Each dataset consists of an equal quantity of interacting pairs and non-interacting pairs. The interacting protein pairs, serving as positive samples, were collected from the public Database of Interacting Proteins (DIPs)[54]. Samples with fewer than 50 amino acids and more than 40% pairwise sequence identity to one another were excluded to reduce fragments and sequence similarity. Negative samples of each dataset were generated by randomly selecting protein pairs in distinct sub-cellular compartments. Proteins from different sub-cellular compartments usually do not interact with each other, and indeed, this construction assures high confidence in identifying negative samples[14].

All additional information of datasets used in this study can be found at the Supplementary Note 4.

**K-embedding strategies**. For a given molecular complex with $m$ molecules, denote $S_m = \{s_1, s_2, \ldots, s_m\}(m \geq 2)$ the set of the corresponding sequences. The set of associated NLP 1-embeddings is $\{\tau_{u_1}^{(1)}(s_1), \tau_{u_2}^{(1)}(s_2), \ldots, \tau_{u_m}^{(1)}(s_m)\}$. Here the subscript ($u_i$) is the embedding dimension, e.g., 512 for the latent space dimension of small molecular Transformer[24]. Our goal is to construct an optimal $m$-embedding model ($\tau_z^{(m)}(S_m)$) from $\{\tau_{u_1}^{(1)}(s_1), \tau_{u_2}^{(1)}(s_2), \ldots, \tau_{u_m}^{(1)}(s_m)\}$, for the complex.

In general, a $q$-embedding is defined on lower forms as the following formula:

$$\tau_w^{(q)}(S_q) := H(\tau_u^{(r)}(S_r), \tau_v^{(t)}(S_t)), \tag{1}$$

where $r + t = q$, and $S_r = \{s_{i_1}, s_{i_2}, \ldots, s_{i_r}\}$, $S_t = \{s_{j_1}, s_{j_2}, \ldots, s_{j_t}\}$, and $S_q = \{s_{k_1}, s_{k_2}, \ldots, s_{k_q}\}$ are three subsets of sequences. Here, the $H$ is the integrating function. In this study, we applied Stack, Prod, and Diff based on the homogeneity or heterogeneity of strategies of lower forms as our choices of $H$.

Specifically, the Stack can be defined as follows:

$$Stack(\tau_u^{(r)}(S_r), \tau_v^{(t)}(S_t)) = \tau_u^{(r)}(S_r) \oplus \tau_v^{(t)}(S_t) \tag{2}$$

where $\oplus$ is the direct sum.

Furthermore, if the lower form strategies are homogenous (i.e., $u = v$, $s = t$), we can define the Prod and Diff as follows:

$$Prod(\tau_u^{(r)}(P_r), \tau_v^{(t)}(P_t)) = \frac{prod - \mu(prod)}{\sigma(prod)}, \tag{3}$$

$$Diff(\tau_u^{(r)}(P_r), \tau_v^{(t)}(P_t)) = \frac{diff - \mu(diff)}{\sigma(diff)}, \tag{4}$$

where $\mu$ and $\sigma$ are the mean value and standard deviation, and

$$prod = \tau_u^{(r)}(P_r) \times \tau_v^{(t)}(P_t), \tag{5}$$

$$diff = \tau_u^{(r)}(P_r) - \tau_v^{(t)}(P_t), \tag{6}$$

where $\times$ and $-$ is the element-wise product and subtraction, respectively.

In this work, the optimization is made over individual NLP embedding ($\tau_{u_j}^{(1)}(s_j)$), such as Transformer, autoencoder, and LSTM, and all the integrating functions ($H$), i.e., Stack, Prod, and Diff.

**Machine learning algorithms**. We use two set of machine learning algorithms. The first set is the artificial neural networks (ANN), a deep learning algorithm that inspired from the complicated functionality of human brain. For each task, we use Bayesian optimization[55] to search the best combination of hyperparameters including network size, L2 penalty parameters, learning rate, batch size, and max iteration. The second model is the gradient boost decision tree (GBDT), one of the most popular ensemble methods. GBDT has the advantages of robustness against overfitting, insensitivity to hyperparameters, effectiveness in the performance, possession of interpretability. GBDT was mainly used to implement regression tasks. The hyperparameters including "n_estimators, max_depth, min_sample_split, subsample, max_features" are chosen based on the data size and embedding dimensions of each task. The Supplementary

Note 3 introduces the optimization strategies used in our study. The detailed settings of hyperparameters are presented in the Supplementary Table 2.

**Bayesian optimization for ANN hyperparameter tuning**. Bayesian optimization is a popular approach to sequentially optimize hyperparameters of machine learning algorithms. The Bayesian optimization is to maximize a black-box function $f(x)$ in a space $\mathcal{S}$:

$$x^* = \arg \max_{x \in \mathcal{S}} f(x), \tag{7}$$

In the hyperparameter optimization, $\mathcal{S}$ can be regarded as the search space of hyperparameters, $x^*$ is the set of optimal hyperparameters, and $f(x)$ is an evaluating metric for machine learning performance.

Given $t$ data points $X_t = (x_1, x_2, \ldots, x_t)$ and their values of evaluating matrics $Y_t = (y_1, y_2, \ldots, y_t)$, Gaussian process can model the landscape of $f$ on the entire space $\mathcal{S}$ by fitting $(X_t, Y_t)$[56]. At any novel point $x$, $f(x)$ is modeled by a Gaussian posterior distribution: $p(f(x)|X_t, Y_t) \sim \mathcal{N}(\mu_t(x), \sigma_t^2(x))$, where $\mu_t(x)$ is mean and $\sigma$ is the standard deviation of $f(x)$ predicted by Gaussian process regression:

$$\mu_t(x) = K(x, X_t)[K(X_t, X_t) + \epsilon_n^2 I]^{-1}Y,$$
$$\sigma_t^2(x) = k(x, x) - K(x, X_t)[K(X_t, X_t) + \epsilon_n^2 I]^{-1}K(x, X_t)^T. \tag{8}$$

Here $k$ is the kernel function, $K(x, X_t)$ is a row vector of kernel evaluations between $x$ and the elements of $X_t$ with $[K(x, X_t)]_i = k(x, x_i)$, and $K(X_t, X_t)$ is the kernel matrix with $[K(X_t, X_t)]_{ij} = k(x_i, x_j)$. $\epsilon_n$ is the noise term, which is learned from the regression.

In Bayesian optimization, both predicted mean and standard deviation are used for the decision making for the next evaluating data point. One can either pick the point maximize the mean values of $f(x)$ for a greedy search, or pick the point with the largest standard deviation to gain new knowledge and improve the Gaussian process accuracy on $f(x)$ landscape. The greedy search may largely maximize $f(x)$ in a few iterations and the exploration of uncertain points can benefit for long-term iterations. To balance such a exploitation-exploration trade-off, an acquisition function, $\alpha(x)$, needs to be picked. The decision for the next evaluating point $x_n$ is picked such that it maximizes the acquisition function

$$x_n = \arg \max_{x \in \mathcal{S}} \alpha(x). \tag{9}$$

In this study, we used the upper confidence bound (UCB) acquisition which can handle the trade-off and it has a fast convergent rate[57] for the black-box optimization.

**Evaluation metrics**. In addition to the evaluation metrics introduced in the Supplementary Note 1 (from the Supplementary Eq. 1 to the Supplementary Eq. 7), R-S scores, R-S index, and modelability index are described below.

**Evaluation metrics: R-S scores**. Residue-similarity (R-S) plot is a new kind of visualization and analysis method that can be applied to an arbitrary number of classes proposed by Hozumi et al.[48]. An R-S plot evaluates each sample of given data by two components, the residue and similarity scores. For given dataset $\{(x_m, y_m)|x_m \in R^N, y_m \in Z_L\}_{m=1}^M$, the residue score and the similarity score of a sample $(x_m, y_m)$ are defined as follows:

$$R_m := R(x_m) = \frac{\sum_{x_j \notin C_l} ||x_m - x_j||}{\max_{x_m \in C_l}(\sum_{x_j \notin C_l} ||x_m - x_j||)}, \tag{10}$$

$$S_m := S(x_m) = \frac{1}{|C_l|} \sum_{x_j \in C_l} \left(1 - \frac{||x_m - x_j||}{d_{\max}}\right), \tag{11}$$

where $l = y_m$, $C_l = \{x_m|y_m = l\}$, and $d_{\max} = \max_{x_i, x_j \in C_l} ||x_i - x_j||$. Note that $0 \leq R_m \leq 1$ and $0 \leq S_m \leq 1$. If a sample is far from other classes, it will have a larger residue score; if a sample is well-clustered, it will have a larger similarity score.

The Class residue index (CRI) and class similarity index (CSI) for the $l$-th class can be defined as $CRI_l = \frac{1}{|C_l|} \sum_m R_m$ and $CSI_l = \frac{1}{|C_l|} \sum_m S_m$. Then the class-independent residue index (RI) and similarity index (SI) can be defined:

$$RI := \frac{1}{L} \sum_l CRI_l, \tag{12}$$

$$SI := \frac{1}{L} \sum_l CSI_l. \tag{13}$$

Then the R-S indices which can give a class-independent evaluation of the deviation R- and S- scores[48] can be defined:

$$RSI := 1 - |RI - SI| \tag{14}$$

Note that RSI range from 0 to 1 and a low RSI indicates a large deviation between the R-score and S-score.

**Evaluation metrics: modelability**. The modelability index is defined independently for classification tasks and regression tasks, namely $MODI_{cl}$ and $MODI_{reg}$, respectively, defined as follows[50,51]:

$$MODI_{cl} = \frac{1}{L}\sum_{i=1}^{L}\frac{N_i}{M_i}, \qquad (15)$$

$$MODI_{reg} = 1 - \frac{1}{M}\sum_{i=1}^{M}\frac{1}{K_i}\sum_{j \in C_i^1}|y_i - y_j|, \qquad (16)$$

where $L$ represents the number of classes, $N_i$ is the count of samples in the $i$-th class whose first nearest neighbor is also in the $i$-th class, $M_i$ is the number of samples in the $i$-th class, $M$ is the total number of samples, $C_i^1$ is the 1-nearest neighbor of $i$-th sample, $K_i$ is the count of samples in $C_i^1$ except the $i$-th sample, and $y_i$ represents the $i$-th samples' normalized label.

**Statistics and reproducibility**. We marked the standard deviation of all our cross-validation results on the Supplementary Table 1. For the reproducibility, the repetitions of our experiments are presented in Supplementary Table 3.

**Reporting summary**. Further information on research design is available in the Nature Portfolio Reporting Summary linked to this article.

## Data availability

All datasets are available at https://weilab.math.msu.edu/DataLibrary/2D/. The Supplementary Data 1 provides .xlsx files for reproducing Figs. 2, 3, 4, and 5.

## Code availability

The source codes are available at https://github.com/WeilabMSU/SVS.

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

## Acknowledgements
This work was supported in part by NIH grants R01GM126189 and R01AI164266, NSF grants DMS-2052983, DMS-1761320, and IIS-1900473, NASA grant 80NSSC21M0023, MSU Foundation, Bristol-Myers Squibb 65109, and Pfizer.

## Author contributions
All authors conceived this work, and contributed to the original draft, review and editing. L.S., H.F., and Y.Q. performed experiments and analyzed data. G.-W.W. provided supervision and resources and acquired funding.

## Competing interests
The authors declare no competing interests.
