## [Peer Review File · Communications Biology]

Reviewers' comments:

Reviewer #1 (Remarks to the Author):

The Authors in this article have performed a relatively comprehensive computational studies of molecular interactions (protein vs. other bioactive molecules) using sequence based representation (instead of main stream 3D representations) and related NLP algorithms. The results show clear advantages of this type of approach over 3D representations in terms of accuracy and data abundance. The work is significant on the correct and innovative path to make significant breakthrough in this field, especially to solve the most of the biological and biochemical problems that 3D information of biomolecules are impossible to accessible. In sum, I recommend the publication of this works.

There are a few comments for minor revisions:

1. sequence based protein modeling is not a completely original idea, the methodology has been studied by a few other groups in the 2000s and 2010s and was referred as "protein language models" mentioned in multiple publications. it is worth to dig into previous exploration using NLP in protein modeling and make some comparison to the current work.
2. computer aided drug discovery should refer as CADD instead of CAGD

Reviewer #2 (Remarks to the Author):

This manuscript introduces a new generation of virtual screening models termed sequence-based virtual screening (SVS), which utilizes advanced natural language processing algorithms and deep K-embedding strategies to encode biomolecular interactions without invoking structure-based docking, which would in general lead to poor ligand-binding prediction. The model was tested on four regression datasets involving protein-ligand binding, protein-protein interaction, protein nucleic acid binding, and ligand targeting protein-protein interactions as well as five classification datasets for the same purpose in different biological species. Their results showed that this method has state-of-the-art performance with potential applications in drug discovery and engineering processes such as predicting novel targets for existing drugs or identifying therapeutic agents more efficiently than traditional methods at lower computational cost. Overall, it is a well-developed article where all the key elements are easily identified and supported with logical flow of ideas.

Comments

1. It is unclear how the authors find the best performing K-embedding strategies for each downstream task.
2. Are there any guidelines for selecting a specific K-embedding strategy for different tasks? For instance, is there any biological rationale behind using a particular K-embedding strategy that results in optimized feature extraction?
3. Based on the results presented in the manuscript, the stacking feature method consistently achieves the best performance. Is it still necessary to use the K-embedding strategy?

Reviewer #3 (Remarks to the Author):

This manuscript by Wei and coworkers presents a sequence-based approach (SVS) and the deep K-embedding strategies for modeling both homogeneous and heterogeneous bio-interactions, such as protein-ligand binding (PL), protein-protein binding (PP), protein-nucleic acid binding (PN), as well as ligand inhibition to protein-protein interactions (iPPI). Two applications are demonstrated,

which are binding affinity prediction and protein-protein interaction identification. Overall, the described research is inspiring, but the manuscript was not prepared in the fashion for general readers of Communications Biology. I also have a number of questions/comments listed below. Thus, I recommend publication in Communications Biology after revisions.

First of all, while sequence-based approaches can be complementary to structure-based approaches. I am not fully convinced that "SVS has the potential to dramatically change the current practice in drug discovery". While this work may show some theoretical success, there may be a long way to go before SVS can demonstrate predictive power for drug discovery, especially before any large-scale experimental validation. There are also examples that a single point mutation can alter or even abandon binding/interactions. I am not sure if SVS or sequence-based models have the resolution comparable to molecular models.

The binding affinity studies (Section 2.2) are clear, but without experimental validation and with relative small datasets, is there any overfitting in the SVS model? Was the SVS model only trained with binding affinity data, or with non-interacting data as well? If only the binding data were used, there may be a high false positive rate when applied to virtual screening tasks. Can SVS predicts non binding or weak binding? Also, is there any imbalance in the PL, PP, PN, and iPPI dataset?

Section 2.3 described the studies to identify protein-protein interactions. It is unclear about the definition of interactions (e.g., actual protein-protein contacts, or involvement in the same biological process). The authors claimed "We applied the SVS method to determine whether a pair of proteins interact or not", which seems to assume proteins from the same species are interacting. The entire section focuses on the "K-embedding strategies", but it is hard to connect to any example protein-protein interactions. I recommend to restructure this section.

The overall methodology is illustrated by Figure 1, but it is unclear how K-embedding helps to connect the diverse molecules in the chemical space. It would be great to have a better illustration. Also, what are the values plotted in Figure 2 panel A. If they are binding affinities (no axis and label are shown), the plot shows a clear clustering of different types of interactions. It seems to suggest some bias in the data set.

Reviewer #4 (Remarks to the Author):

Guowei et al present in this manuscript that a K-embedding strategy to combine the pre-trained language models of molecules (proteins, nucleic acids, or small molecules) for inferring their interactions. Extensive validations were presented, including ligand-protein interactions, protein-protein interactions (PPIs), protein- nucleic acid interactions, and interestingly 3-body interactions of ligand targeting PPIs. Three different embedding operations (Stack, Prod, and Diff) were tested, and the Stack operation yields - unsurprisingly as it doesn't reduce dimensions - best results. The study, however, suffers from major flaws that make it unsuitable for publication in its current form.

1) Most of the benchmark results clearly suffers from data leakage, which means that the high correlation of prediction comes from seeing similar sequences during the training, instead of actually learning the interaction pattern, as nicely pointed out by a recent preprint (<https://doi.org/10.1101/2023.01.18.524543>). In fact, the PPI sets, training procedures and testing methods used in this manuscript echoes the five sets in the preprint, it is thus urged that the authors use the new data-leakage-free data set that minimizing sequence similarities between training and test provided in the provided preprint.

The fact that the deep learning models are not learning the actual interactions can be further seen in the PPI inhibitor case (Figure 3). As long as Transformer embedding is used, the correlation is more than 0.76 if only ligand information is provided, and is 0.766 if ligand/protein/protein information are all fed. This indicates that the model is not learning how a ligand is interacting with the interface between two proteins, as such a small difference on the performance can easily comes from the difference in the number of weights, their initial values and associated training practices.

2) Set the data leakage issue aside, the manuscript fails to provide enough merits in why and how K-embedding works. As the underlying sequence models are already well-established and taken directly by this work, and the downstream learning algorithms involved in this work are also commonly available, the major question this paper tried to address is how to centralize the interactive information through embedding, however, few insights are provided on how this is done and why K-embedding is the way to go. To this end, the manuscript provides little for the community.

A minor point at line 54: It's CADD not CAGD.

Dear Reviewers,

Thank you very much for your comments and suggestions on our paper (COMMSBIO-23-0118-T) entitled “SVSBI: Sequence-based virtual screening of biomolecular interactions”.

We have carefully gone through your comments, and have accordingly revised the manuscript. Our specific responses to the comments, and changes made in the manuscript are enclosed.

Reviewer 1

The Authors in this article have performed a relatively comprehensive computational studies of molecular interactions (protein vs. other bioactive molecules) using sequence based representation (instead of main stream 3D representations) and related NLP algorithms. The results show clear advantages of this type of approach over 3D representations in terms of accuracy and data abundance. The work is significant on the correct and innovative path to make significant breakthrough in this field, especially to solve the most of the biological and biochemical problems that 3D information of biomolecules are impossible to accessible. In sum, I recommend the publication of this works.

Answer: We thank the reviewer for a summary of our manuscript, which reveals the core idea of our study. Your positive comments are inspirational and motivates us for further research. Also, we thank the reviewer's suggestions. We have made various changes accordingly.

Comment 1: *sequence based protein modeling is not a completely original idea, the methodology has been studied by a few other groups in the 2000s and 2010s and was referred as "protein language models" mentioned in multiple publications. it is worth to dig into previous exploration using NLP in protein modeling and make some comparison to the current work.*

Answer: We have revised the Introduction (page 2, line 67-68) of sequence based modeling to include previous explorations of NLP model on protein modeling. A few new references are cited.

Comment 2: *computer aided drug discovery should refer as CADD instead of CAGD.*

Answer: The misspell has been revised (page 2, line 53).

Reviewer 2

This manuscript introduces a new generation of virtual screening models termed sequence-based virtual screening (SVS), which utilizes advanced natural language processing algorithms and deep K-embedding strategies to encode biomolecular interactions without invoking structure-based docking, which would in general lead to poor ligand-binding prediction. The model was tested on four regression datasets involving protein-ligand binding, protein-protein interaction, protein nucleic acid binding, and ligand targeting protein-protein interactions as well as five classification datasets for the same purpose in different biological species. Their results showed that this method has state-of-the-art performance with potential applications in drug discovery and engineering processes such as predicting novel targets for existing drugs or identifying therapeutic agents more efficiently than traditional methods at lower computational cost. Overall, it is a well-developed article where all the key elements are easily identified and supported with logical flow of ideas.

Answer: We thank the reviewers for the positive comments. We have also addressed the concerns below.

Comment 1: *It is unclear how the authors find the best performing K-embedding strategies for each downstream task.*

Answer: We have modified our version to emphasize the techniques for searching the optimal K-embedding strategies in Section 2.1 (page 5, line 139-141).

Comment 2: *Are there any guidelines for selecting a specific K-embedding strategy for different tasks? For instance, is there any biological rationale behind using a particular K-embedding strategy that results in optimized feature extraction?*

Answer: For different tasks, computationally, we used Bayesian optimization or grid search for searching the optimal K-embedding strategy via the best combination of hyperparameters. For the second question, there are some biological rationales that might affect the effectiveness of a particular K-embedding strategy. For instance, for PPIs, the prediction results obtained by using the stack strategy will depend on the order of two input sequences. Therefore, biases caused by the input order of proteins in the training set may be systematically captured by the machine learning model, causing interference in the prediction of interactions. Additionally, for different biological interactions, the importance of the sequence information of the molecules involved may be different. And different K-embedding can effectively manifest the importance of the sequence information of specific molecules, avoiding the model from focusing on specific molecules, which might lead to overfitting.

Comment 3: *Based on the results presented in the manuscript, the stacking feature*

method consistently achieves the best performance. Is it still necessary to use the K-embedding strategy?

Answer: Before training on the dataset, we could not determine whether there was a systematic bias in the input order of a type of particular molecule in the dataset, nor the importance of a particular type of molecules in this effect. Therefore, searching the k-embedding strategies can ensure our model attain the optimal performance. In addition, considering that other strategies are still effective for the prediction of molecular interactions where their results are at the similar level of stack strategy, these strategies may give the optimal prediction results in unknown task scenarios.

Reviewer 3

This manuscript by Wei and coworkers presents a sequence-based approach (SVS) and the deep K-embedding strategies for modeling both homogeneous and heterogeneous bio-interactions, such as protein-ligand binding (PL), protein-protein binding (PP), protein-nucleic acid binding (PN), as well as ligand inhibition to protein-protein interactions (iPPI). Two applications are demonstrated, which are binding affinity prediction and protein-protein interaction identification. Overall, the described research is inspiring, but the manuscript was not prepared in the fashion for general readers of Communications Biology. I also have a number of questions/comments listed below. Thus, I recommend publication in Communications Biology after revisions.

Answer: We thank the reviewers for the positive comments. The concerns are addressed below.

Comment 1 : *First of all, while sequence-based approaches can be complementary to structure-based approaches. I am not fully convinced that "SVS has the potential to dramatically change the current practice in drug discovery". While this work may show some theoretical success, there may be a long way to go before SVS can demonstrate predictive power for drug discovery, especially before any large-scale experimental validation. There are also examples that a single point mutation can alter or even abandon binding/interactions. I am not sure if SVS or sequence-based models have the resolution comparable to molecular models.*

Answer: Thank you for raising this important question. We are sorry that our description of this idea is absolute in the original manuscript. The description has been revised (page 1, line 22). For predicting biological mutation, related works have been done recently (see <https://doi.org/10.1101/2022.12.18.520933> and <https://doi.org/10.1038/s43588-021-00168-y>). These two works suggest that sequence-based models are feasible for predicting the functional changes induced by single or multiple mutations.

Comment 2: *The binding affinity studies (Section 2.2) are clear, but without experimental validation and with relative small datasets, is there any overfitting in the SVS model? Was the SVS model only trained with binding affinity data, or with non-interacting data as well? If only the binding data were used, there may be a high false positive rate when applied to virtual screening tasks. Can SVS predicts non binding or weak binding? Also, is there any imbalance in the PL, PP, PN, and iPPI dataset?*

Answer: 1. It is all known that overfitted models do not have predictive power for cross-validation. We did four binding affinity datasets, namely PL, PP, PN, and iPPI. PL dataset is the benchmark dataset PDBbind version 2016, which has been used in many previous studies, consisting of a given training dataset and an independent test

dataset. PP and PN datasets are generated from PDBbind version 2020. A 10-fold-cross-validation with 20 repetitions was taken for these datasets, which shows the free of overfitting. The iPPI dataset is taken from Rodrigues et al. The results involved a 10-fold-cross-validation with 50 repetitions (details are available in the Supplement). We followed the standard training-test splitting procedure, and detected no overfitting of our models.

2. For binding affinity studies, our SVS model was trained on interacting data. Also, our SVS model did work effectively on non-interacting data of protein- protein interaction, which can be found in Section 2.3. And our results show that SVS are robust in predicting binding or non binding. Thus, we believe that the SVS model can be applied to virtual screening scenarios through the following two steps: ① predicting binding or non binding targets; and ② for each binding target, predicting its binding affinity.

3. We have included the imbalance analysis in Section 3. In conclusion, no imbalance is detected for PL, PP, and PN datasets. However, for iPPI dataset, imbalance exists as this datasets consist 1694 ligands but only 31 PPIs since this dataset is directly constructed based on 31 target-dependent binding affinity prediction tasks. We have included in the discussion of such imbalance and its affection on our SVS model in Section 2.2.4 (page 10, line 287-300).

***Comment 3:** Section 2.3 described the studies to identify protein-protein interactions. It is unclear about the definition of interactions (e.g., actual protein-protein contacts, or involvement in the same biological process). The authors claimed “We applied the SVS method to determine whether a pair of proteins interact or not”, which seems to assume proteins from the same species are interacting. The entire section focuses on the “K-embedding strategies”, but it is hard to connect to any example protein-protein interactions. I recommend to restructure this section.*

Answer: Thank you for your constructive suggestion regarding the definition of interaction. We have revised Section 2.3 (page 10-12, line 307-310). We have added one paragraph in Section 3 about why K -embeddings are chosen in our studies(Page 12-13, line 343-351). We believe that with the addition of the above content, the K -embedding strategy in Section 2.3 will be more natural and easy to understand.

***Comment 4:** The overall methodology is illustrated by Figure 1, but it is unclear how K -embedding helps to connect the diverse molecules in the chemical space. It would be great to have a better illustration. Also, what are the values plotted in Figure 2 panel A. If they are binding affinities (no axis and label are shown), the plot shows a clear clustering of different types of interactions. It seems to suggest some bias in the data set.*

Answer: Thank you for your reasonable and important comment. We have revised caption of Figure 1 to give a better explanation of K -embedding and how it works on diverse molecular datasets. Additionally, for Figure 2 panel A, there is no bias in the data set as the x - and y - axis do not represent the actual value of binding affinities. Each dataset is scaled to a specific interval range for the clarity of visualization. We have revised the axes to improve the clarity.

Reviewer 4

Guowei et al present in this manuscript that a K-embedding strategy to combine the pre-trained language models of molecules (proteins, nucleic acids, or small molecules) for inferring their interactions. Extensive validations were presented, including ligand-protein interactions, protein-protein interactions (PPIs), protein- nucleic acid interactions, and interestingly 3-body interactions of ligand targeting PPIs. Three different embedding operations (Stack, Prod, and Diff) were tested, and the Stack operation yields - unsurprisingly as it doesn't reduce dimensions - best results. The study, however, suffers from major flaws that make it unsuitable for publication in its current form.

Answer: We thank the reviewer for the important suggestions and comments. We have addressed the concerns in the revision.

Comment 1: *Most of the benchmark results clearly suffers from data leakage, which means that the high correlation of prediction comes from seeing similar sequences during the training, instead of actually learning the interaction pattern, as nicely pointed out by a recent preprint (<https://doi.org/10.1101/2023.01.18.524543>). In fact, the PPI sets, training procedures and testing methods used in this manuscript echoes the five sets in the preprint, it is thus urged that the authors use the new data-leakage-free data set that minimizing sequence similarities between training and test provided in the provided preprint.*

The fact that the deep learning models are not learning the actual interactions can be further seen in the PPI inhibitor case (Figure 3). As long as Transformer embedding is used, the correlation is more than 0.76 if only ligand information is provided, and is 0.766 if ligand/protein/protein information are all fed. This indicates that the model is not learning how a ligand is interacting with the interface between two proteins, as such a small difference on the performance can easily comes from the difference in the number of weights, their initial values and associated training practices.

Answer: Thank you very much for your enlightening suggestion. We have carefully read the preprint (<https://doi.org/10.1101/2023.01.18.524543>), and understand that it is indeed a serious issue in general. In our case, we just adopt benchmark datasets and follow the standard training-test splitting to compare with the literature. A dedicated study of data leakage is out the scope of our work. However, we have added discussions about data leakage in Section 3 (page 15-16, line 431-439).

In general, machine learning models, including deep learning models, are data-driven approaches that rely on learning from data. If the similarity of train data and test data is minimizing, any data-driven model would treat them as objects in different categories and thus undoubtedly give random predictions. However, although our model does rely

on the similarities of feature representations and data distributions, we have ample evidences that our model does not rely on molecules’ sequence similarity. First of all, the actual input of our model is the K -embeddings generated from NLP models. Some of K -embeddings will alter the similarity of original input (i.e., NLP embeddings). Furthermore, many studies suggest important information, such as the structure information, contextual information, and evolutionary information, of a biomolecule can be directly capture from its sequences by NLP models. Thus, even if two molecules share similar sequences, their NLP embedding may have great differences. Based on this idea, many researches have been proposed and achieved great success for studying protein engineering (see <https://doi.org/10.1101/2022.12.18.520933> and <https://doi.org/10.1038/s43588-021-00168-y>).

Finally, for the PPI inhibitor task, we never assume that our SVS model did fully learn the actual interactions in this specific task as there are apparently a data imbalance issue as we discussed in Section 2.2.4 (page 10, line 287-300). In this task, the most important comparison object of our method are the traditional ligand-based (i.e., target-dependent) methods. Those methods are theoretically incapable of capturing the actual interactions. However, by introducing the K -embedding strategies, SVS can study interactions involved multiple molecules, enabling learning the actual interactions of PPI inhibition theoretically feasible. Therefore, our primary goal here is to demonstrate that SVS can learn as much interactive information as possible from extremely limited PPI information. And this task is done based on a 10-fold-cross-validation with 50 repetitions. All parameters of the model are systematically and automatically optimized by our preset scheme (all our other studies using the same scheme) as we discussed in Section 2.1. Furthermore, different NLP models and K -embedding strategies are taken and they all show similar conclusions. Therefore, the difference in the results shown in our manuscript is probably difficult to obtain with pure training. Furthermore, the example that K -embedding generated from One-hot PPI pair provides another case that our model does not reply on the sequence similarity. One-hot PPI pair embedding only describe whether two PPIs are in the same PPI family or not. In this case, our model can only know whether different PPI targets are similar or not. Our results show that the PPI target similarity information has very limited improvement compared with some other K -embeddings. This shows that sequence similarity information is not very helpful for SVS to predict the PPI inhibition.

***Comment 2** Set the data leakage issue aside, the manuscript fails to provide enough merits in why and how K -embedding works. As the underlying sequence models are already well-established and taken directly by this work, and the downstream learning algorithms involved in this work are also commonly available, the major question this paper tried to address is how to centralize the interactive information through embedding,*

however, few insights are provided on how this is done and why K-embedding is the way to go. To this end, the manuscript provides little for the community.

Answer: Thank you very much for raising this point. We have added one paragraph about why K -embeddings are chosen (page 12-13, line 343-351). We regard the traditional single-molecule language model as the basis, inductively define high order K -embedding, and provide a systematic framework which could be directly applied to any molecule language models for arbitrarily many-molecule-interaction scenarios. Admittedly, some parts of our algorithm can be considered as “commonly available”. Nevertheless, these parts are just some special cases described in our framework. We integrate and promote some effective strategies in traditional research, resulting in the definition of K -embedding, so that SVS can be naturally applied in areas that have not been explored before. To our best knowledge, we are the first to consider PPI inhibition as an interaction involved 3-body molecules and conduct research through a sequence-based model. Our work demonstrates the importance and generality of our K -embeddings for unknown research tasks.

Comment 3 *A minor point at line 54: It's CADD not CAGD.*

Answer: We have revised this misspell (page 2, line 53).

REVIEWERS' COMMENTS:

Reviewer #2 (Remarks to the Author):

The authors have successfully addressed my comments.

Reviewer #4 (Remarks to the Author):

I'm pleased with the additional comments and explanations provided, and believe that the manuscript, in its current form, will serve as a valuable contribution to the scientific community.